# Assessing the effectiveness of intervention to prevent plague through community and animal-based survey

**Soanandrasana Rahelinirina**[1]*, **Soloandry Rahajandraibe**[2], **Sitraka Rakotosamimanana**[3], **Minoarisoa Rajerison**[1]

**1** Plague Unit, Institut Pasteur de Madagascar, Antananarivo, Madagascar, **2** Central Laboratory for Plague, Ministry of Public Health, Antananarivo, Madagascar, **3** Epidemiology Unit, Institut Pasteur de Madagascar, Antananarivo, Madagascar

* rahelinirina@pasteur.mg

## Abstract

Bubonic plague, transmitted by infected flea bites, is the most common form of plague and, left untreated, can progress to the pneumonic form, which is highly contagious. Surveillance focusing on reservoir and vector is considered to be the main approach to prevent plague. Common rodent control methods include the use of rodenticide and snap traps but, in a plague context, the dispersal of fleas from killed animals may pose a serious health threat. Therefore, there is a need for strategies which address reservoir and vector control. The aim of this study was to assess the effects of combination of reservoir and vector control through community-based surveillance. Activities were implemented by local previously trained community agents in two active plague foci in Madagascar. Kartman bait stations containing rodenticide and insecticide were placed indoors while live traps were set outdoors. Small mammals were identified and killed with their fleas. Effectiveness of control measures was evaluated by comparison of plague incidence two years before and after intervention using data on reported human cases of plague from the Central Laboratory of Plague. A total of 4,302 small mammals were captured, with the predominance of the black rat *Rattus rattus*. Our results found a reduction in plague incidence in the treated site for at least two years after treatment. Community-based interventions played an important role in reducing contact between humans-rodents-fleas. Our study confirms the importance of animal surveillance during the low plague transmission season. The combination of reservoir and vector control with community involvement may be effective at reducing the risks of plague spillover to humans. The strategy of using Kartman bait stations indoors with live traps outdoors can be used to refine proactive plague prevention, however, due to the potential development of resistance to pesticides in flea and rat populations, overuse should be considered.

**Data Availability Statement:** The authors confirm that the data supporting the findings of this study

are available within the article and its supplementary files.

**Funding:** The Fondation de Lille and Institut Pasteur de Madagascar. Grant number : No. 14-001/2016. The funders had no role in study design, data collection and analysis, decision to publish, or preparation of the manuscript.

**Competing interests:** The authors have declared that no competing interests exist.

## Introduction

Plague, a zoonotic disease caused by the bacterium *Yersinia pestis*, remains a major public health threat in Madagascar, which reported 80.5% of all human cases in the world between 2015 and 2018 [1]. Bubonic plague, resulting from an infected flea bite, is the most common form and, left untreated, can progress to the pneumonic form, which is highly contagious through human-to-human transmission.

In Madagascar, the main reservoir of plague is the black rat (*Rattus rattus*), which is widely distributed throughout the country. Two flea species are reported to be vectors of plague; *Xenopsylla cheopis*, the oriental rat flea which displays a worldwide distribution and is mainly living on indoor rats, and *Synopsyllus fonquerniei*, an endemic flea of rats living outdoors [2, 3]. In the Central Highlands, where most human plague cases occur every year, plague season runs mainly from October to April [4], but some cases are sometimes observed as early as August [5, 6]. The abundance of rodent populations in rural highlands corresponds to the season of low plague transmission and is related to crop harvesting. However, the abundance of flea vector populations corresponding to the season of high plague transmission is mainly linked to temperature and humidity. Considering the rising incidence of endemic plague in rural areas, it appears urgent to develop effective measures against plague infection that could target all possible routes of transmission as well as disease persistence.

One of the most common approaches is to implement community-based interventions to reduce the abundance of rats and their fleas both indoors and outdoors in order to reduce the risks of pathogen spillover to humans. Rodent control usually relies on the use of acute rodenticide and snap traps [7] but in the context of plague, the dispersal of fleas from killed rats may pose a serious health threat.

The aim of this study was to implement an efficient strategy to prevent human plague cases using animal-based surveillance and local community involvement. Evidence of the effectiveness of this strategy was evaluated by comparing human morbidity due to plague before and after intervention in rural plague endemic foci.

## Materials and methods

### Ethics statement

Before the onset of the intervention by itself, the local communities were informed of the purpose of the study and verbal consent was obtained. All animal capture and handling procedures were done in accordance with directive 2010/63/EU of the European Parliament and of the Council (https://eur-lex.europa.eu/LexUriServ/LexUriServ.do?uri=OJ:L:2010:276:0033:0079:en:PDF) as well as the guidelines of the American Society of Mammologists for the use of wild animals in research and education [8].

Plague is a notifiable disease, and the data used for this study were gathered from the National Plague Control Program (NPCP) in Madagascar with mandatory notification at the central level of all suspected plague patients; no ethics approval was required. In addition, information for all human cases has been anonymized before data sharing for analysis. The authors did not have access to any information that could identify individual participants during or after data collection.

### Study sites

The study was performed in Tsiroanomandidy district within three rural Communes, Bemahatazana, Miandrarivo and Ambatolampy, in the Midwest parts of Madagascar (with intervention) and in Ambositra district located in the southern central highlands (without

intervention), approximately 100 km from Tsiroanomandidy ([Fig 1]). These two districts were chosen due to their high number of bubonic plague cases every year [9, 10].

### Field experiment

The specific interventions that were implemented consisted in a combination of both rodent and vector control coupled with Information, Education and Communication (IEC) activities. All were conducted during the low plague season from April to September 2016. Rodent control consisted of both live capture and killing via poison. Kartman bait stations [11], which combine rodent and vector controls, and wire mesh live traps, hereafter designed as to BTS (Besançon Technique Service) [3] were used and monitored by local community agents during the whole study. The Kartman bait station is a wooden box with a lid that limits access by nontarget species and children. Interventions were conducted in 112 villages and hamlets, after obtaining the verbal consent of the householders. Per village, 20 to 100 Kartman bait stations containing 15g of bait block anticoagulant rodenticide (Chlorophacinone, BHL, Madagascar) and 150g of insecticide powder fenitrothion 2% (an insecticide used in the NPCP) per bait station were set inside households (one bait station per house). Rodents enter the station, eat the rodenticide bait, soak their fur with the insecticide powder that they bring back to their nest where they are expected to die within four days. Every day, bait stations were checked, baits were replenished and bait consumption was recorded (bait either consumed or not).

Twenty (20) BTS live traps were set outside each village, being spaced 10m from each other and baited with dried fish and onion. Interventions were conducted for 5 days/4 nights every month in each village (6 times per village). Kartman bait station and BTS were set each evening and checked the next morning. The captured animals were identified to species on morphological grounds. Sex and age were not recorded. Each live-captured animal was humanely killed [8], then immediately incinerated and buried with its fleas.

Three hundred fact sheets and 112 booklets that explain plague transmission routes, symptoms, treatment and prevention, produced by the Madagascar Ministry of Public Health and other partners, were distributed and posted in a public place in each village. In addition, IEC was raised on methods of rat proofing, impacts of bush fires and waste management. Fifty-nine local community agents and healthcare workers were recruited and trained to conduct these interventions during the six months of the study.

### Temporal trend of plague incidence

In order to evaluate the effectiveness of the community-based intervention, we compared the incidence of human plague cases between the treatment (Tsiroanomandidy) and the control (Ambositra) sites. Plague is a disease requiring mandatory notification and all data on notified plague cases in Madagascar are centralized in a database at the Central Plague Laboratory hosted at the Institut Pasteur de Madagascar. The patients were not recruited specially for the study, they are part of mandatory notification of plague case by health centers. Anonymized data from August 2014 to July 2018 ([S1] and [S2] Tables) were used to construct the deviation from the triennial mean of the incidence according to WHO's plague case definition [12].

### Data analysis

Statistical analyses were carried out using Microsoft Excel and Stata 13 software. The calculation of the trend in deviations from the 5-year mean incidence or incidence fluctuation was adapted from the calculation of the Deviation from decadal mean of incidence (DDMI) performed by Rakotosamimanana *et al*. [10]. Based on the estimated population of the district for the study period, we calculated the mean monthly incidence per year of probable and

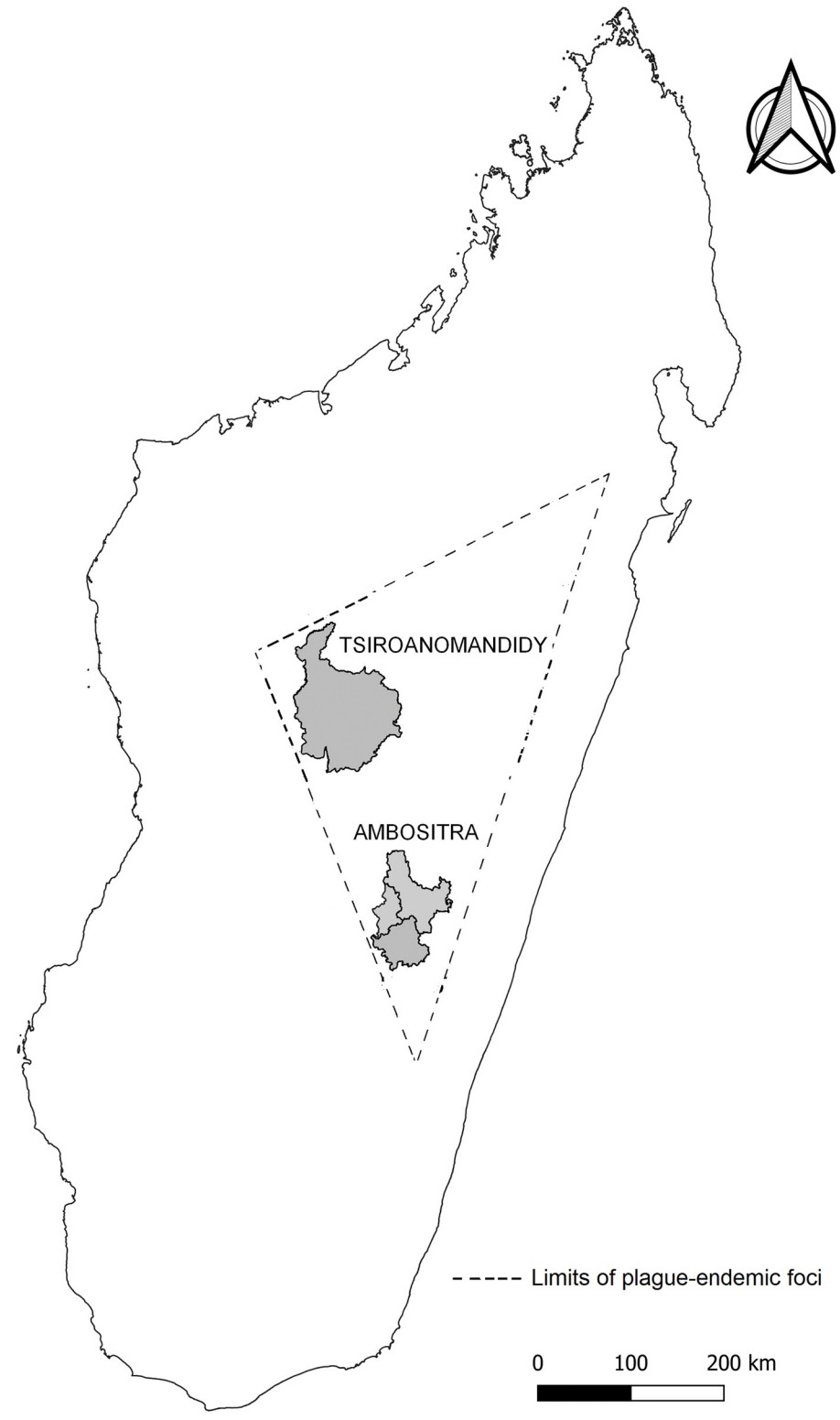

**Fig 1. Map of Madagascar showing the studied districts and the limits of plague-endemic foci.** The dashed line represents the limits of the endemic foci of plague. Performed by S. Rahajandraibe using free and open-source Quantum GIS (QGIS) 2.8 software. Shapefile source for administrative limits: BNGRC (National Disaster Management Office), polygons cleaned and merged by UNOCHA (United Nations Office for the Coordination of Humanitarian Affairs, https://www.unocha.org/) in December 2017: All maps are in the public domain, (https://data.humdata.org/dataset/cod-ab-mdg).

confirmed human plague cases. Then, we calculated the mean plague incidence for the whole study period. Finally, the trend of the deviations of the mean incidence for the 5-year studied period was analysed with a linear regression line of equation $y = ax+b$.

Small mammal relative abundance was estimated using trap success, defined as the ratio of trapped individuals to trap night. Rodenticide bait consumption was noted as consumed (whatever partially or totally) or not consumed each trapping night.

## Results and discussion

In total, 4,302 small mammals were captured with live traps set outside among 16,800 trap nights during the six months of the study with average trap success of 25.6%. Two species were captured *R. rattus* (4130; 96%) and *Suncus murinus* (172; 4%). For each village, trap success varied from 14.1% to 52.6%. Our monthly data revealed that there was no significant decrease in trap success from April to September (Table 1). In particular, the relative abundance of small mammals did not reach a peak in the harvest crop seasons (June and July), as observed in a previous study [13]. This may be explained by compensatory immigration and/or reproduction.

Looking at the Kartman bait stations, within 75,764 observations, rodenticide bait consumption varied for each house, but bait consumption was higher during the two first months of intervention and then decreased during the four last months (Fig 2). We did not find rat carcasses probably because the intoxicated animals expectedly died inside their nests.

Fig 3 shows a fluctuating plague incidences trend over 4-year in both locations. A marked decreased trend was found in Tsiroanomandidy after the intervention as well as one year later (Fig 3A). Furthermore, plague cases started later in the next plague season, starting in October instead of August. In Ambositra, an increased trend was observed across the same time period (Fig 3B).

Our study is the first one in Madagascar to demonstrate that rodent and flea control, both indoor and outdoor coupled with IEC activities during the season of low plague transmission may be efficient to reduce incidence of human plague by decreasing both reservoir and vector populations. The low plague season corresponds to the cold and dry season, while the highest abundance of rodents coincides with the crop harvest period [13]. Flea abundance is low in periods of cold temperatures and low humidity [3]. A previous study showed that, in Tsiroanomandidy, human cases occur every year and, in 2014, *Y. pestis* was isolated from *R. rattus* [14]. Our study shows that *R. rattus* indeed remains the predominant small mammals in this rural area. The use of the Kartman bait stations during the low plague season seems to decrease the flea population, hence the risk of flea dispersal compared to the high plague season which the abundance of fleas is high. The use of anticoagulant rodenticide has the advantage of killing rodents within a few days after ingesting a lethal dose, thus leaving time for the insecticide to

**Table 1. Monthly trapping success of small mammals over the whole sampling device and period.**

| Session | Trap nights (TN) | Nb of small mammals captured (N) | Trap success N*100/TN (%) |
|---|---|---|---|
| April | 2,702 | 518 | 19.2 |
| May | 2,808 | 693 | 24.7 |
| June | 2,822 | 1,007 | 35.7 |
| July | 2,823 | 831 | 29.4 |
| August | 2,815 | 678 | 24.1 |
| September | 2,830 | 575 | 20.3 |
| **Total** | **16,800** | **4,302** | **25.6** |

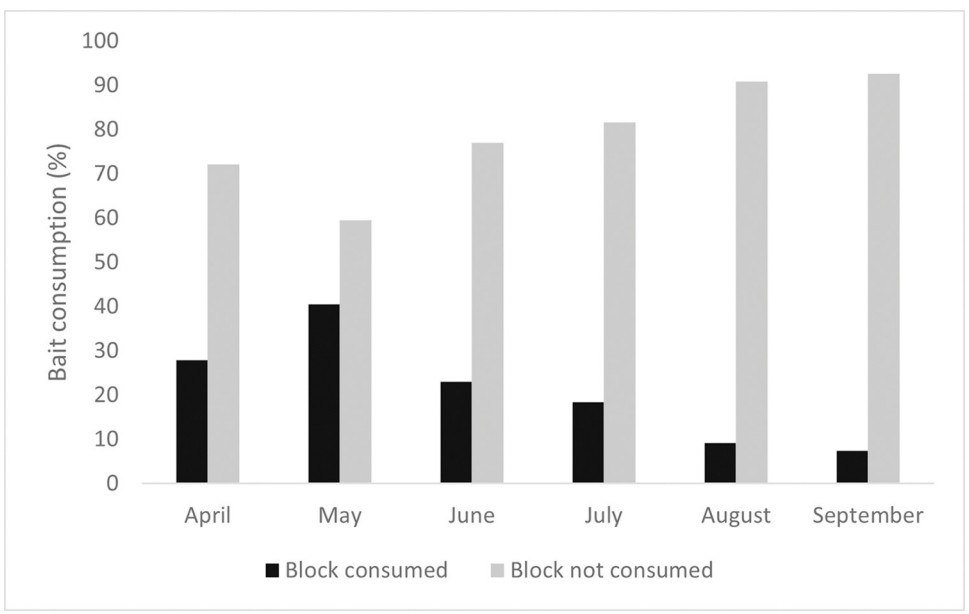

**Fig 2. Percentage of rodenticide bait consumption based on bait consumed (black) or not consumed (grey).**

work and kill the fleas before the rodents die, and thus prevent their dispersion. Over the last five years, the same experiences using Kartman bait station were conducted in other active Malagasy plague foci, and similar results were obtained, with decreasing rodent population [15], thus suggesting that the community-based strategy presented here can be successfully extended to other settings.

A previous study showed that intervention targeting only inside homes was not sufficient to reduce rodent population abundance, probably because of rapid recolonization by outdoor rat [15]. Here, we show that regular rodent control both indoors and outdoors is efficient, as it may prevent rodents to re invade home too rapidly. However, to undertake successful rodent and vector control, accurate data on the biology and behavior of rodent species as well as good knowledge of the associated environmental conditions and risks are necessary prior the implementation [16]. For instance, outdoor rodenticide treatment is usually not advised due to its deleterious impact on environment and non-target species including human [17]. Kartman bait station appears beneficial as it delivers both the rodenticide and insecticide simultaneously, thus reducing both rodent and flea populations. However, whilst insecticide and rodenticide have been shown to be effective, their sustained use may trigger the development of resistance mechanisms [18, 19]. Choosing an appropriate combination of rodenticide and insecticide for use in a national program against plague is an important prerequisite for effective interventions. In Madagascar, interventions using animal-based surveillance are rarely implemented except when investigating a recent or ongoing outbreak. The use of Kartman bait stations fueled with insecticide with bait without rodenticide is recommended as a reactive action during epidemic situations in order to target the fleas before they bite humans and to avoid potential flea release from dead rodents. During plague inter-seasons, rodent and flea controls could be implemented before the beginning of the human plague season at the village level as an additional preventive action. However, healthcare workers alone may not be sufficient to implement such a resource-demanding strategy. The involvement of the local communities is thus necessary. In Tsiroanomandidy, bush fires are commonly used to produce green growth for livestock and to prepare agricultural fields at the end of the dry season (August and

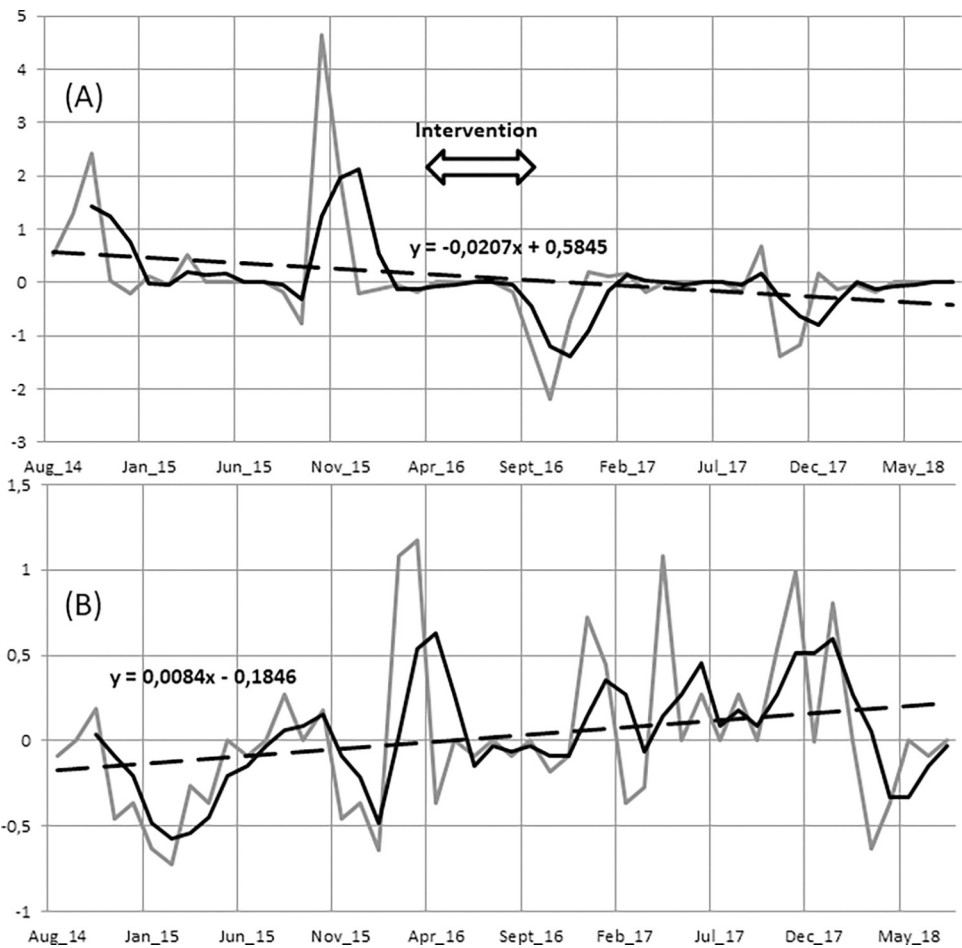

**Fig 3.** Deviation from the quadrennial mean of the incidence (per 100,000 inhabitants) from August 2014 to July 2018 in Tsiroanomandidy with intervention (A) and Ambositra without intervention (B). The grey curve represents the deviation from the quadrennial mean of the incidence (DQMI); the black curve represents the quarterly moving averages of the DQMI; the dashed line is a linear regression line showing the trend of the DQMI for the study period.

September). This period coincides with the end of the low plague season. Fire is expected to have an impact on the habitat of rodent predators such as birds or snakes, and it may induce rodent dispersal towards human habitats, and increase rodent-human interactions. Bush fires can also exacerbate climate change which promotes growth and development of fleas. Educating local communities needs to achieve a balance between advantages and disadvantages of using fire [20]. Radio and television play a role in the communication and education of people, but given the low resources of the population of the endemic plague areas, they are not easily and widely accessible. Therefore, community-based sensitization and fact sheets illustrated in the local language probably constitute the most valuable tools to communicate about disease risk. Due to the potential development of insecticide resistance in flea populations [18], the regular assessment of the status of fleas to insecticide must also be determined. As surveillance of rodent and flea populations across the entire endemic plague foci is costly, relying of generalized community-led surveys and trapping campaigns could be a simple and cheap approach to manage rodents both indoors and outdoors.

The limitations of this study are the lack of data for the flea index, which could be useful to complete the effectiveness of the use of insecticide in the Kartman bait stations, as well as the

rather simplistic measurement of rodent activities towards the poison-containing baits. However, local population feedbacks appear to confirm that the use of Kartman bait station within households did prove efficient to reduce rodent abundance.

## Conclusions

Our study confirms the importance of preventive program during low plague transmission to reduce indoor and outdoor small mammal abundance in order to avoid subsequent human-rodent-flea interactions during the following plague season. We thus suggest that prioritizing such cheap and simple prevention should be considered as a very effective way to fight plague, especially given the high financial costs when outbreaks occur.

Our strategy of using Kartman bait stations indoors together with live traps outdoor can be used to refine proactive plague prevention. However, due to the potential development of insecticide and rodenticide resistance in fleas and rodents, the use of various pesticides is recommended. Our finding can serve as a guide for determining when and how rodent control should be done in order to reduce plague incidence in rural areas. Long-term actions focusing on education, hygiene around villages, including reducing refuse and cover, appropriate food storage and rat proofing are complementary strategies that would probably help to plague diminution. Decentralization of health services could also improve access to health care and dissemination of knowledge about plague-associated risks.

## Supporting information

**S1 Table. The triennial mean of the incidence from August 2014 to July 2018 in Tsiroanomandidy (treated site).**
(XLSX)

**S2 Table. The triennial mean of the incidence from August 2014 to July 2018 in Ambositra (control site).**
(XLSX)

## Acknowledgments

I would like to thank all those that participated in this study especially the community health workers and health care agents in the district of Tsiroanomandidy, the population participating in Bemahatazana, Miandrarivo and Ambatolampy, the staff of Plague Unit and Parfait Andriniaina.

## Author Contributions

**Conceptualization:** Soanandrasana Rahelinirina, Minoarisoa Rajerison.

**Data curation:** Soanandrasana Rahelinirina, Soloandry Rahajandraibe, Sitraka Rakotosamimanana.

**Formal analysis:** Soanandrasana Rahelinirina, Soloandry Rahajandraibe, Sitraka Rakotosamimanana.

**Funding acquisition:** Minoarisoa Rajerison.

**Investigation:** Soanandrasana Rahelinirina, Soloandry Rahajandraibe, Minoarisoa Rajerison.

**Methodology:** Soanandrasana Rahelinirina, Sitraka Rakotosamimanana, Minoarisoa Rajerison.

**Project administration:** Soanandrasana Rahelinirina, Minoarisoa Rajerison.

**Resources:** Minoarisoa Rajerison.

**Software:** Sitraka Rakotosamimanana.

**Supervision:** Soanandrasana Rahelinirina, Minoarisoa Rajerison.

**Validation:** Soanandrasana Rahelinirina, Minoarisoa Rajerison.

**Visualization:** Soanandrasana Rahelinirina, Minoarisoa Rajerison.

**Writing – original draft:** Soanandrasana Rahelinirina.

**Writing – review & editing:** Soanandrasana Rahelinirina, Soloandry Rahajandraibe, Sitraka Rakotosamimanana, Minoarisoa Rajerison.

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
