## [Decision Letter · Decision Letter 0]

3 Aug 2023

PGPH-D-23-00799

Assessing the effectiveness of intervention to prevent plague through community and animal-based survey

Dear Dr. Soanandrasana Rahelinirina

Thank you for submitting your manuscript to PLOS Global Public Health. After careful consideration, we feel that it has merit but does not fully meet PLOS Global Public Health’s publication criteria as it currently stands. Therefore, we invite you to submit a revised version of the manuscript that addresses the points raised during the review process.

omments to author:

Reviewer 1

Decision : acceptance of the manuscript.

Reviewer 2

Minor revision

Comments

The authors demonstrate the importance of animal surveillance in areas of low plague prevalence. And two ways to refine proactive plague prevention, indoor and outdoor, are proposed. This strategy is economic and effective. The author's results and exploration of prevention are also very meaningful for reducing plague epidemics and preventing plague epidemics in other parts of the world.

Since the author's work was carried out in 2016, it is recommended to add some progress of related work in recent years to the discussion.

Reviewer 3

Major revision

Comments

The authors have presented only detailed analysis of the human data. There is very little analysis of the rodent trapping data which would be more informative to a wider audience interested in the rodent side of the study as well as the human incidence of plague infection.

Details of the rodent trapping data are not available.

English expression requires considerable improvement. See attached pdf.

We look forward to receiving your revised manuscript.

Kind regards,

Mohamed Salah Abbassi

Academic Editor

Journal Requirements:

Additional Editor Comments (if provided):

Reviewers' comments:

Reviewer's Responses to Questions

**Comments to the Author**

1. Does this manuscript meet PLOS Global Public Health’s publication criteria? Is the manuscript technically sound, and do the data support the conclusions? The manuscript must describe methodologically and ethically rigorous research with conclusions that are appropriately drawn based on the data presented.

Reviewer #1: Yes

Reviewer #2: Yes

Reviewer #3: Partly

2. Has the statistical analysis been performed appropriately and rigorously?

Reviewer #1: Yes

Reviewer #2: Yes

Reviewer #3: No

3. Have the authors made all data underlying the findings in their manuscript fully available (please refer to the Data Availability Statement at the start of the manuscript PDF file)?

Reviewer #1: Yes

Reviewer #2: Yes

Reviewer #3: No

4. Is the manuscript presented in an intelligible fashion and written in standard English?

Reviewer #1: Yes

Reviewer #2: Yes

Reviewer #3: No

5. Review Comments to the Author

Reviewer #1: The authors bring in the manuscript results from Assessing the effectiveness of an intervention to prevent plague through community and animal-based surveys. The approach is quite complete. The surveillance the authors report on has valuable information to share. It is good to report surveillance results in known endemic regions to determine risks to plague. Therefore, the publication is important.

Reviewer #2: The authors demonstrate the importance of animal surveillance in areas of low plague prevalence. And two ways to refine proactive plague prevention, indoor and outdoor, are proposed. This strategy is economic and effective. The author's results and exploration of prevention are also very meaningful for reducing plague epidemics and preventing plague epidemics in other parts of the world. Since the author's work was carried out in 2016, it is recommended to add some progress of related work in recent years to the discussion.

Reviewer #3: The authors have presented only detailed analysis of the human data. There is very little analysis of the rodent trapping data which would be more informative to a wider audience interested in the rodent side of the study as well as the human incidence of plague infection.

Details of the rodent trapping data are not available.

English expression requires considerable improvement. See attached pdf.

6. PLOS authors have the option to publish the peer review history of their article (what does this mean?). If published, this will include your full peer review and any attached files.

**Do you want your identity to be public for this peer review?** For information about this choice, including consent withdrawal, please see our Privacy Policy.

Reviewer #1: **Yes: **Saber Esmaeili

Reviewer #2: **Yes: **Xin Wang

Reviewer #3: No

---

## [Editor Report · Decision Letter 1]

26 Oct 2023

PGPH-D-23-00799R1

Assessing the effectiveness of intervention to prevent plague through community and animal-based survey

Dear Dr. Rahelinirina,

Thank you for submitting your manuscript to PLOS Global Public Health. After careful consideration, we feel that it has merit but does not fully meet PLOS Global Public Health’s publication criteria as it currently stands. Therefore, we invite you to submit a revised version of the manuscript that addresses the points raised during the review process.

We look forward to receiving your revised manuscript.

Kind regards,

Mohamed Salah Abbassi

Academic Editor

Journal Requirements:

Additional Editor Comments (if provided):

PGPH-D-23-00799

Assessing the effectiveness of intervention to prevent plague through community and animal-based survey

Dear Dr. Soanandrasana Rahelinirina

Thank you for submitting your manuscript to PLOS Global Public Health. After careful consideration, we feel that it has merit but does not fully meet PLOS Global Public Health’s publication criteria as it currently stands. Therefore, we invite you to submit a revised version of the manuscript that addresses the points raised during the review process.

omments to author:

Reviewer 1

Decision : acceptance of the manuscript.

Reviewer 2

Minor revision

Comments

The authors demonstrate the importance of animal surveillance in areas of low plague prevalence. And two ways to refine proactive plague prevention, indoor and outdoor, are proposed. This strategy is economic and effective. The author's results and exploration of prevention are also very meaningful for reducing plague epidemics and preventing plague epidemics in other parts of the world.

Since the author's work was carried out in 2016, it is recommended to add some progress of related work in recent years to the discussion.

Reviewer 3

Major revision

Comments

The authors have presented only detailed analysis of the human data. There is very little analysis of the rodent trapping data which would be more informative to a wider audience interested in the rodent side of the study as well as the human incidence of plague infection.

Details of the rodent trapping data are not available.

English expression requires considerable improvement. See attached pdf.

A rebuttal letter that responds to each point raised by the editor and reviewer(s). You should upload this letter as a separate file labeled 'Response to Reviewers'.

We look forward to receiving your revised manuscript.

Kind regards,

Mohamed Salah Abbassi

Academic Editor

Journal Requirements:

Additional Editor Comments (if provided):

Reviewers' comments:

Reviewer's Responses to Questions

Comments to the Author

1. Does this manuscript meet PLOS Global Public Health’s publication criteria? Is the manuscript technically sound, and do the data support the conclusions? The manuscript must describe methodologically and ethically rigorous research with conclusions that are appropriately drawn based on the data presented.

Reviewer #1: Yes

Reviewer #2: Yes

Reviewer #3: Partly

2. Has the statistical analysis been performed appropriately and rigorously?

Reviewer #1: Yes

Reviewer #2: Yes

Reviewer #3: No

3. Have the authors made all data underlying the findings in their manuscript fully available (please refer to the Data Availability Statement at the start of the manuscript PDF file)?

Reviewer #1: Yes

Reviewer #2: Yes

Reviewer #3: No

4. Is the manuscript presented in an intelligible fashion and written in standard English?

Reviewer #1: Yes

Reviewer #2: Yes

Reviewer #3: No

5. Review Comments to the Author

Reviewer #1: The authors bring in the manuscript results from Assessing the effectiveness of an intervention to prevent plague through community and animal-based surveys. The approach is quite complete. The surveillance the authors report on has valuable information to share. It is good to report surveillance results in known endemic regions to determine risks to plague. Therefore, the publication is important.

Reviewer #2: The authors demonstrate the importance of animal surveillance in areas of low plague prevalence. And two ways to refine proactive plague prevention, indoor and outdoor, are proposed. This strategy is economic and effective. The author's results and exploration of prevention are also very meaningful for reducing plague epidemics and preventing plague epidemics in other parts of the world. Since the author's work was carried out in 2016, it is recommended to add some progress of related work in recent years to the discussion.

Reviewer #3: The authors have presented only detailed analysis of the human data. There is very little analysis of the rodent trapping data which would be more informative to a wider audience interested in the rodent side of the study as well as the human incidence of plague infection.

Details of the rodent trapping data are not available.

English expression requires considerable improvement. See attached pdf.

6. PLOS authors have the option to publish the peer review history of their article (what does this mean?). If published, this will include your full peer review and any attached files.

Do you want your identity to be public for this peer review? For information about this choice, including consent withdrawal, please see our Privacy Policy.

Reviewer #1: Yes: Saber Esmaeili

Reviewer #2: Yes: Xin Wang

Reviewer #3: No
---

## [Editor Report · Decision Letter 2]

20 Nov 2023

Assessing the effectiveness of intervention to prevent plague through community and animal-based survey

PGPH-D-23-00799R2

Dear authors

We are pleased to inform you that your manuscript 'Assessing the effectiveness of intervention to prevent plague through community and animal-based survey' has been provisionally accepted for publication in PLOS Global Public Health.

Best regards,

Mohamed Salah Abbassi

Academic Editor

Dear authors

Thanks you for submiting your article to Plos Global Public Health. I reach a decision to accept your article.

Sincerely